# Conformal Prediction for Generative Models via Adaptive Cluster-Based Density Estimation

## Abstract

Conditional generative models map input variables to complex, high-dimensional distributions, enabling realistic sample generation in a diverse set of domains. A critical challenge with these models is the absence of calibrated uncertainty, which undermines trust in individual outputs for high-stakes applications. To address this issue, we propose a systematic conformal prediction approach tailored to conditional generative models, leveraging density estimation on model-generated samples. We introduce a novel method called CP4Gen, which utilizes cluster-based density estimation to construct prediction sets that are less sensitive to outliers, more interpretable, and of lower structural complexity than existing methods. Extensive experiments on synthetic datasets and real-world applications, including climate emulation tasks, demonstrate that CP4Gen consistently achieves superior performance in terms of prediction set volume and structural simplicity. Our approach offers practitioners a powerful tool for uncertainty estimation associated with conditional generative models, particularly in scenarios demanding rigorous and interpretable prediction sets.

## 1 Introduction

Modern machine learning applications increasingly rely on generative models to produce complex, high-dimensional outputs such as images, text, and molecular structures (Karras et al., 2019; Brown et al., 2020; Gómez-Bombarelli et al., 2018). These models, such as variational autoencoders (Kingma & Welling, 2022), generative adversarial networks (Goodfellow et al., 2014), and diffusion models (Song et al., 2021), have shown impressive performance across domains ranging from natural language processing to scientific discovery. However, a critical limitation of these generative models is their lack of calibrated uncertainty estimates (Gal & Ghahramani, 2016). Although they can generate realistic samples, it remains unclear how much we should trust a single output, particularly in high-stakes domains like medical diagnosis or autonomous systems.

Conformal prediction (CP) provides a model-agnostic, post-hoc framework that yields prediction sets with finite-sample coverage guarantees while maintaining sharpness (Vovk et al., 2005; Angelopoulos et al., 2025). However, classical conformal prediction methods are primarily designed for regression and classification tasks. Given the need of uncertainty quantification for generative models, a conformal prediction extension is desired. There have recently been some initial attempts to extend conformal prediction to the generative setting. For example, Wang et al. (2022) proposed probabilistic conformal prediction (PCP), which constructs prediction sets as unions of balls centered at ensemble members from conditional generative models. While PCP is broadly applicable in the sample-only setting, its isotropic, structure-free construction tends to over-cover and produce disconnected components sensitive to outliers.

This work proposes a new CP method, called CP4Gen, which successfully addresses these limitations. It introduces a lightweight structural assumption on ensemble members (i.e., Gaussian mixture) through clustering and local covariance estimation, enabling prediction sets that are sharper, more robust, and more interpretable. This approach adopts the split conformal prediction framework (Lei et al., 2017), which avoids the high computational cost of leave-one-out CP by splitting the dataset into training and calibration sets. In this paper, we also provide a density estimation view of CP for generative models. It unifies CP4Gen and PCP under the same framework and outlines design space for more generative CP methods.

(a) Calibration of CP4Gen

(b) Inference of CP4Gen

Figure 1: **The two phases of CP4Gen: calibration and inference.** (a) $K$-means is applied to prediction ensemble members. Each identified cluster is treated as one mode of a Gaussian Mixture Model, whose sample mean $\boldsymbol{\mu}_k$ and covariance matrix $\boldsymbol{\Sigma}_k$ are calculated. Nonconformity score $s_i$ is defined as the minimal $s_i^k$ between observation $Y_i$ and each cluster. Nonconformity score quantile $Q_{1-\alpha}$ is then computed on the calibration set. (b) Given a new set of covariates, ensemble members are sampled and $K$-means is applied on top. Prediction set is obtained by inverting nonconformity score function with $Q_{1-\alpha}$ resulting as a union of $K$ ellipsoids.

Our main contributions are summarized as follows:

- We formulate an extension of the conformal prediction framework to the generative setting. This new approach works with any existing conditional generative models and only requires the capability to draw samples from the models.

- We define a new evaluation metric for generative model prediction sets, called structural complexity. Structural complexity measures the prediction set's geometric intricacy and reflects how easily the prediction set can be interpreted and used for decision-making.

- We propose a new CP method, CP4Gen, following our approach. CP4Gen introduces a tunable hyper-parameter which controls resulting set's structural complexity, rendering PCP as one special case.

- Evaluated on a variety of datasets, we show that CP4Gen outperforms state-of-the-art PCP in terms of prediction set volume and structural complexity, and consistently demonstrates superior performance on responses of various dimensionality.

## 2 Related Works

**Conformal Prediction with Deterministic Predictions** Conformal prediction was conventionally designed for deterministic models, which produce a single regression prediction for each input. Literature explicitly leveraging output geometry structure to construct prediction sets is extensive. For instance,

Johnstone & Cox (2021); Messoudi et al. (2022) approximate output residual distribution with a Gaussian model, resulting in ellipsoidal prediction sets. Messoudi et al. (2021) uses copulas to capture non-linear dependencies across output dimensions, and Tumu et al. (2024) constructs prediction regions using a union of convex templates for multi-modal coverage. However, these methods produce global prediction sets that do not adapt to input variations. Another line of approaches leverages transport maps to construct prediction sets. Essentially, these methods transform a complex residual distribution into a simple reference distribution, so that well-defined quantile sets from the reference distribution can be pulled back to form conformal prediction sets in the original output space. Klein et al. (2025); Thurin et al. (2025) build such transport maps using optimal transport, and Feldman et al. (2023); Fang et al. (2024) learn them using generative models, such as variational autoencoders (Kingma & Welling, 2022) and normalizing flows (Rezende & Mohamed, 2015).

**Conformal Prediction with Conditional Distribution Predictions** In many settings, predictive models output a full conditional distribution given the input. A common approach is to design a family of nested sets on the output distributions and then select the smallest sets within the family that achieves the desired marginal coverage. In multiclass classification, where models output a probability mass function over a finite label set, CP methods return a prediction set consisting of all labels whose total probability exceeds a calibrated threshold (Sadinle et al., 2019). In the case of univariate continuous response, Chernozhukov et al. (2021) and Sesia & Romano (2021) use estimated conditional cumulative distribution functions to produce valid prediction intervals. In multivariate continuous settings, CD-split (Izbicki et al., 2019) constructs prediction sets by thresholding an estimated conditional density, and Izbicki et al. (2021) refine CD-split to improve conditional coverage.

**Conformal Prediction with Generative Models** Modern generative models fall into two classes: those with tractable conditional densities and those with intractable conditional densities. The tractable class, including normalizing flows (Rezende & Mohamed, 2015) and flow matching models (Lipman et al., 2024), permits point-wise evaluation of conditional density functions, and the conformal methods with distribution predictions apply. The intractable class, including score-based diffusion models (Ho et al., 2020; Song et al., 2021) and other implicit samplers, lies between the tractable generative models and deterministic predictors by supplying an ensemble of finite samples per input, yet do not provide an explicit conditional density. It positions us in a new regime where previous conformal methods are not applicable, and motivates the development of new methods that can exploit ensembles to efficiently calibrate uncertainty over generated samples.

## 3 Methodology

### 3.1 Problem Setup

Consider a dataset consisting of $N$ i.i.d. sample pairs of covariates $X_i$ and target variables $Y_i$, denoted as $\mathbf{D} = \{(X_i, Y_i)\}_{i=1}^N$. Each sample pair $(X_i, Y_i)$ is generated from some unknown joint distribution $P_{XY}$. When the covariates $X_{N+1}$ of a new data point is given, the goal is to construct a prediction set $\hat{C}_\alpha(X_{N+1})$ for the unobserved target $Y_{N+1}$ with valid uncertainty estimation using the dataset $\mathbf{D}$. Specifically, the prediction set $\hat{C}_\alpha(\cdot)$ should satisfy the following criteria:

$$P_{XY}(Y_{N+1} \in \hat{C}_\alpha(X_{N+1})) \geq 1 - \alpha, \tag{1}$$

for any $\alpha \in [0, 1]$. Obviously, an arbitarily large prediction set can trivially satisfy the above criteria. We thus require the prediction set to be as small as possible.

Classic conformal prediction uses leave-one-out estimation (Vovk et al., 2005), which has high computational cost due to repeated prediction model fitting. Our work is instead based on the split conformal prediction framework (Lei et al., 2017; Papadopoulos et al., 2002), which significantly reduces the computational cost by splitting the dataset and only fitting the prediction model once. Specifically, the dataset $\mathbf{D}$ is randomly split into a preliminary set $\mathbf{D}_p$ and a calibration set $\mathbf{D}_c$. The prediction model is fit on $\mathbf{D}_p$, then kept fixed to compute the nonconformity scores on $\mathbf{D}_c$ and to predict at a new data point $X_{N+1}$.

### 3.2 Generative Modeling

In our setup, the prediction model is a conditional generative model $q(Y|X)$ that approximates the target variable conditional distribution $P_{Y|X}$. The prediction set is then constructed based on the random samples drawn from the generative model given a new set of covariates as input. This setup is the same as the one in Wang et al. (2022) and Zheng & Zhu (2024). Note that it is a different approach than the conventional conformal prediction methods that are based on summary statistics of the target distribution such as the mean and quantiles (Lei et al., 2017; Romano et al., 2019) and rely on evaluating the whole probability densities of the target distribution (Chernozhukov et al., 2021; Hoff, 2021; Izbicki et al., 2019).

The simple prerequisite of having samples from a target distribution allows our method to be compatible with all the popular generative models, such as variational autoencoders (Kingma & Welling, 2022), generative adversarial networks (Goodfellow et al., 2014), and flow-based models (Lipman et al., 2024). It also makes our method applicable to a wide range of real-world applications, where generative modeling is well-suited, such as weather forecasting (Price et al., 2025; Yang et al., 2025), climate downscaling (Yang et al., 2022; 2024a; Harder et al., 2023), natural hazards mitigation (Giezendanner et al., 2023; Saunders et al., 2025), sequence modeling (Yang et al., 2024b), protein structure prediction (Watson et al., 2023), and image generation (Karras et al., 2022).

In this work, we demonstrate our method on various target conditional distribution uncertainty estimation tasks, using a flow-matching model – a popular conditional generative model (model details in Appendix B.2).

### 3.3 The Density Estimation View of Conformal Prediction

Given a preliminary set $\mathbf{D}_p$, a conditional generative model $q(Y|X)$ can be trained to approximate the target conditional distribution $P_{Y|X}$. For a data point $(X_i, Y_i)$ in the calibration set $\mathbf{D}_c$, one may define a simple nonconformity score as:

$$s(X_i, Y_i|q) = -\log q(Y_i|X_i), \tag{2}$$

where $s(\cdot|q): \mathcal{X} \times \mathcal{Y} \to \mathbb{R}$ is the nonconformity score function associated with the generative model $q$. $s(\cdot|q)$ is an intuitive choice of score function for a generative model, as it measures the discrepancy between the target variable observation and the estimated conditional distribution. Specifically, a small score $s(X_i, Y_i|q)$, i.e., a large $q(Y_i|X_i)$, indicates that the target variable $Y_i$ is well-explained by the generative model $q$ and thus conforms well to the estimated conditional distribution $q(Y|X_i)$.

The empirical quantile of the nonconformity scores on the calibration set $\mathbf{D}_c$ is then computed to construct the prediction set for new data points. The $\alpha$-th quantile of the nonconformity scores is given by:

$$Q_\alpha(\{s(X_i, Y_i|q)\}_{i=1}^N) = \inf_x \left\{ \frac{1}{N} \sum_{i=1}^N \mathbb{I}\left(s\left(X_i, Y_i|q\right) \leq x\right) \geq \alpha \right\}, \tag{3}$$

where $\alpha \in [0, 1]$, $\mathbb{I}(\cdot)$ is the indicator function, and $N$ is the number of samples in the calibration set $\mathbf{D}_c$. Suppose that the desired nominal coverage is $1 - \alpha$. Given a new data point $X_{N+1}$, the prediction set is then constructed by:

$$\hat{C}_\alpha(X_{N+1}) = \left\{ Y : s(X_{N+1}, Y|q) \leq Q_{1-\alpha}\left(\{s\left(X_i, Y_i|q\right)\}_{i=1}^N \cup \{\infty\}\right) \right\}. \tag{4}$$

However, for most generative models, we have no closed-form expression for $q(Y|X_i)$. Instead, we only have access to samples drawn from $q(Y|X_i)$ through the generative model. Assume we have $M$ random samples, $\hat{Y}_i^1, \ldots, \hat{Y}_i^M$, independently generated from $q(Y|X_i)$, denoted as $\hat{\mathbf{Y}}_i = \{\hat{Y}_i^1, \ldots, \hat{Y}_i^M\}$. Then the problem boils down to estimating the conditional density function of $q(Y|X_i)$ from empirical samples $\hat{\mathbf{Y}}_i$, and characterizing its level sets to construct prediction regions. While density estimation has been well-studied through methods such as kernel density estimation and finite mixture modeling (Rosenblatt, 1956; Parzen, 1962; McLachlan & Peel, 2000), tracking density level sets introduces additional challenges, as discussed in the following sections.

### 3.4 Instantiations

Varying how we estimate the density function of $q(Y|X_i)$ leads to different nonconformity score function designs, and thus different conformal prediction methods. In this section, we first instantiate a specific density estimator and build its connection to an existing conformal prediction method. We then identify key limitations of this approach and propose a more efficient alternative.

#### 3.4.1 Probabilistic Conformal Prediction (PCP)

PCP (Wang et al., 2022) is a conformal prediction method whose nonconformity score is defined as the minimum distance from $Y_i$ to the ensemble samples $\hat{Y}_i^m$ (Equation 5c). This leads to a prediction set consisting of $M$ balls, each with the same radius and centered at an ensemble prediction. Within our proposed framework, where the nonconformity score arises from sample-based density estimation, PCP can be viewed as approximating the ensemble distribution with a Gaussian mixture model (GMM) of $M$ modes. Each Gaussian component is centered at a generated sample $\hat{Y}_i^m$ and uses the same diagonal covariance matrix $\Sigma = \sigma^2 I$. Its nonconformity score can be rewritten as:

$$s(X_i, Y_i|\hat{q}) = -\log \sum_{m=1}^{M} \frac{1}{M} \, \mathcal{N}(Y_i; \hat{Y}_i^m, \sigma^2 I) \tag{5a}$$

$$\approx -\log \max_{1 \leq m \leq M} \frac{1}{M} \, \mathcal{N}(Y_i; \hat{Y}_i^m, \sigma^2 I) \tag{5b}$$

$$\overset{\text{rank}}{\sim} \min_{1 \leq m \leq M} \left\| Y_i - \hat{Y}_i^m \right\|. \tag{5c}$$

Note the GMM density is approximated by the largest mode (Equation 5b). It leads to a closed-form expression for the resulting prediction set, which can be computed efficiently as a union of $M$ convex sets.

#### 3.4.2 Conformal Prediction for Generative Models (CP4Gen)

One obvious drawback of PCP is that its prediction sets fail to concentrate on high-density regions. PCP constructs the prediction set as a union of $M$ balls, of which each has the same radius. Since the balls are uniformly sized regardless of local density, balls centered on outliers inflate the set into low probability regions, wasting volume, while balls in dense regions may inadequately capture the local structure. In addition, the prediction set structural complexity (i.e., the number of Gaussian modes) is $M$, which makes it impractical to use when $M$ is large. More descriptions on structural complexity are provided in Section 4.1.

In light of the above drawbacks, we propose a new conformal prediction method for generative models denoted as CP4Gen, which is more robust to outliers and has lower structural complexity. A visual overview of the proposed method is provided in Figure 1. CP4Gen first fits $K$-means on the $M$ generated outputs. It treats each found cluster as a mode of a Gaussian mixture model, and computes the cluster's sample mean $\boldsymbol{\mu}_k$, covariance $\boldsymbol{\Sigma}_k$, and weight $w_k$ (i.e., the proportion of samples in the $k$-th cluster). This leads to the following nonconformity score function:

$$s(X_i, Y_i|\hat{q}) = -\log \sum_{k=1}^{K} w_k \, \mathcal{N}(Y_i; \boldsymbol{\mu}_k, \boldsymbol{\Sigma}_k) \tag{6}$$

$$\approx -\log \max_{1 \leq k \leq K} w_k \, \mathcal{N}(Y_i; \boldsymbol{\mu}_k, \boldsymbol{\Sigma}_k + \beta^2 I),$$

where $\beta^2 I$ is a small constant diagonal matrix that ensures the covariance matrix is positive-definite and can be removed when $K$ is relatively small compared to $M$. $K$ is a hyperparameter that controls the model's sensitivity to outliers and the resulting prediction set's structural complexity. Note that when $K = M$, CP4Gen becomes PCP, where each ensemble prediction is regarded a cluster. In practice, the hyperparameter $K$ can be fine-tuned on $\mathbf{D}_p$ to balance prediction set volume and structural complexity. To keep the tuning cost low, $K$ is searched on a coarse grid. In our following experiments, $K$ is optimized to minimize the prediction set volume on $\mathbf{D}_p$. The $K$ corresponding to the smallest volume is used for the testing stage.

Algorithm 1 summarizes all the details for applying Equation equation 6 to construct prediction sets. In this algorithm, we use $K$-means clustering with Euclidean distance as a pre-processing step for fitting a Gaussian mixture model of the conditional distribution. It yields centroids equal to the local mean, ensuring that the clusters are consistent with the downstream density modeling. We also experimented with expectation maximization (EM) for Gaussian mixture fitting, but EM has more computation overhead, and the final performance is similar to the $K$-means one.

### 3.5  Analysis

As presented in Section 3.4, both PCP and CP4Gen first fit a conditional density using a prediction ensemble, and then compute nonconformity score from the fit. They differ in how the conditional density is estimated: PCP employs a kernel density estimator with an undesirably small bandwidth $\sigma^2$, such that the estimated density remains biased for all $M$, whereas CP4Gen uses a Gaussian mixture model, yielding a more consistent density estimate. When the conditional distribution is an input-ignorant Gaussian distribution, CP4Gen produces a conformal set that converges to the highest-density set of the true Gaussian distribution and is asymptotically sharper than PCP's. This result is formalized in Proposition 3.1, with the proof provided in Appendix F.

**Proposition 3.1.** *Assume the true conditional distribution $P(Y|X)$ is uni-mode Gaussian with mean $\mu$ and covariance matrix $\Sigma$, and the generative model outputs i.i.d. ensemble data from $P(Y|X)$, then the conformal set given by CP4Gen gets asymptotically sharper as $M$ increases, while the set given by PCP does not.*

A similar argument extends to the case where true conditional distribution is a mixture of well-separated Gaussians. For more general non-Gaussian targets, both PCP and CP4Gen suffer from biased density estimates. However, CP4Gen can flexibly adjust its expressivity through the choice of $K$, leading to reduced bias and tighter prediction sets.

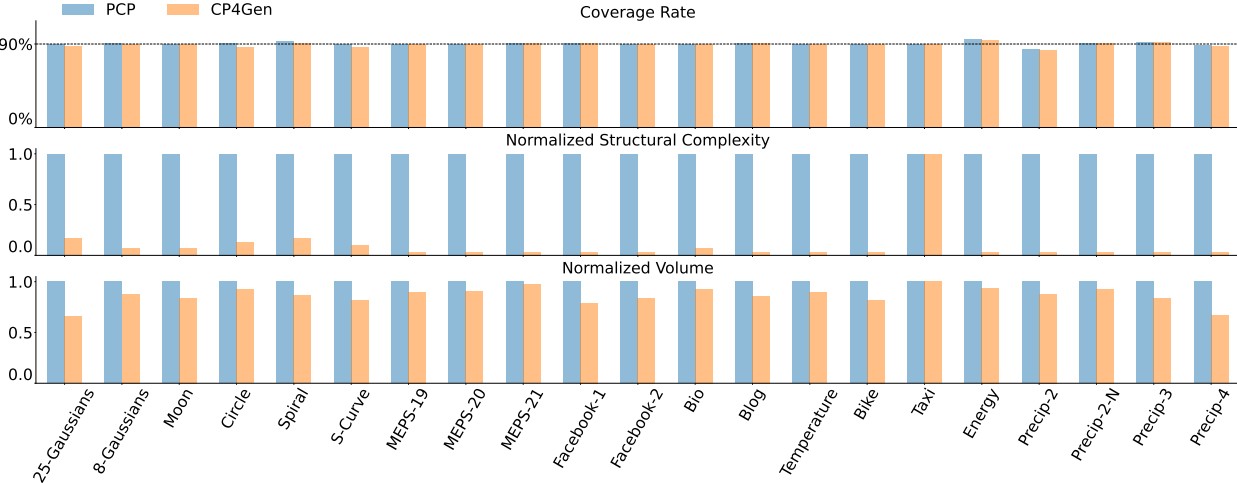

Figure 2: **A comprehensive performance summary of CP methods.** CP methods ($\alpha = 0.1$) are extensively evaluated on various datasets including synthetic, real-world, and precipitation emulation tasks. The performance of PCP and CP4Gen are compared in terms of prediction set coverage rate (a), structural complexity (b), and volume (c). The results of structural complexity and volume are normalized to $[0, 1]$ for visualization convenience. Results in original scale are reported in the appendix Tables 1, 2, and 3. Both CP methods achieve desired coverage rate close to $1 - \alpha = 0.9$ on all datasets. CP4Gen consistently outperforms PCP in terms of both prediction set volume and structural complexity. In particular, the CP4Gen produced prediction sets' structural complexity is significantly lower than PCP's, reducing complexity by more than $90\%$ on most datasets.

## 4 Experiments

### 4.1 Evaluation Metrics

We evaluate the conformal prediction set based on the following three metrics: empirical coverage rate, prediction set volume, and prediction set structural complexity.

**Empirical Coverage Rate** Conformal prediction methods guarantee a marginal coverage rate in theory under the exchangeability assumption. Here we use the empirical coverage rate to examine how well this theoretical guarantee holds in practice. In this work, we set the significance level $\alpha = 0.1$ for all the experiments and therefore expect an empirical coverage rate close to 90%.

**Prediction Set Volume** In addition to ensuring valid coverage, efficiency is another key evaluation metric of CP performance. It is assessed by the volume of the prediction set $|\hat{C}_\alpha(X_{N+1})|$. While an arbitrarily large prediction set would guarantee coverage, it offers little practical value for decision-making. As a result, a prediction set is optimal if it minimizes the volume while maintaining the coverage. The set volume can be computed exactly in 1-d response space (i.e. the length of intervals). In higher dimensions, we approximate the volume with Monte Carlo sampling.

**Prediction Set Structural Complexity** Besides the two commonly used metrics, coverage and volume, we also consider the structural complexity of the prediction set. It is defined as the number of unique convex sets whose union is the prediction set. It measures the shape complexity of prediction sets and tells us how easily the prediction set can be interpreted and used for downstream decision-making. For example, it is more informative and actionable for a wind farm operator to know that next hour average wind speed falls within the range $[10, 20]$ m/s than within $[10, 14.01] \cup [14.05, 16.08] \cup [17, 19.32] \cup [19.5, 20]$ m/s. It would be unclear whether the multi-modal nature of the second set is statistically significant or just a spurious consequence of overfitting. Also, when prediction sets are used for risk assessment and robust optimization in high-stakes settings, one must identify the worst/best case point within the prediction region. The optimization overhead would be much higher on complex sets than on simple ones. In the case of convex optimization, the time complexity scales linearly with the number of convex sets (Johnstone & Cox, 2021; Bertsimas et al., 2011). In our wind example, the second set would take 4 times longer to optimize than the first set.

### 4.2 Computational Complexity

CP4Gen is a computationally efficient conformal prediction method. All steps are tractable in high dimensions: $K$-means clustering is operated in low-dimensional distance metrics; both evaluating and inverting the Gaussian distribution density have analytic solutions. Therefore, CP4Gen can be applied to high-dimensional response spaces with moderate computational overhead. The resulting prediction set is the union of several ellipsoids, whose volume does not have a closed-form expression. It needs to be estimated by Monte Carlo sampling which suffers from curse of dimensionality. However, it is only necessary when evaluating CP4Gen performance in terms of sharpness.

The main computational complexity of CP4Gen comes from inverting the estimated covariance matrix associated with each cluster. In high-dimensional settings, inverting a full covariance matrix is computationally expensive and numerically unstable, especially when the ensemble size is small compared to dimensionality $d$. A natural remedy is to impose a diagonal or low-rank-plus-diagonal structure on the covariance matrix, which leads to a more efficient inverse computation. While some expressive power is lost relative to a full covariance, the reduction in prediction set volume of CP4Gen is largely driven by the mixture decomposition and conditional scoring rather than the exact covariance parameterization. As a result, diagonal or low-rank covariances will retain strong volume reduction benefits while ensuring computational feasibility in high dimensions.

### 4.3 Datasets

We compare CP4Gen to the state-of-the-art CP methods PCP and HD-PCP (Wang et al., 2022) on various synthetic and real-world datasets. HD-PCP requires additional per-sample confidence scores, making it a unfair comparison against CP4Gen. Thus, we save this comparison in Appendix D.

**One-dimensional Response** The synthetic datasets include classic 2-d data such as `25-Gaussians`, `8-Gaussians`, `Moon`, `Circle`, `Spiral`, and `S-Curve`, where one dimension is the input and the other is the response. Real data experiments are conducted on nine public-domain datasets: bike sharing data (`Bike`), physicochemical properties of protein tertiary structure (`Bio`), blog feedback (`Blog`), and Facebook comment volume, variants one (`Facebook-1`) and two (`Facebook-2`), medical expenditure panel survey number 19 (`MEPS-19`), number 20 (`MEPS-20`), and number 21 (`MEPS-21`) (Romano et al., 2019), and temperature forecast data (`Temperature`). Experiments on these nine datasets take in multi-dimensional covariates and output a single response (c.f. Appendix B.1 for further details on the datasets). These nine public-domain datasets are the same ones which were used in Wang et al. (2022) for evaluation of PCP. They are structurally diverse and represent a variety of domains including physics, biology, and social science. We here use the same datasets to permit a proper benchmarking of CP methods in the context of generative models.

**Two-dimensional Response** Beyond single-response tasks, we also study the multi-response generation setting, where the response is a vector of multiple variables. For this setting, we consider two real-world datasets: `Taxi` and `Energy`. The `Taxi` (Kaggle, 2021) dataset consists of New York City taxi trip records which include the pickup, drop-off locations of each trip and the corresponding time. We randomly sample 10000 records from January 2016 (6000, 2000, 2000 for train, calibration and test respectively), and use the pickup time and location as covariates to predict drop-off locations. The locations are represented by latitude and longitude pairs. For the `Energy` dataset, we predict the heating load and cooling load for energy efficiency analysis (Tsanas & Xifara, 2012). The predictors are building information such as orientation, glazing area, and wall area. We use 460, 154, and 154 samples for train, calibration and test set.

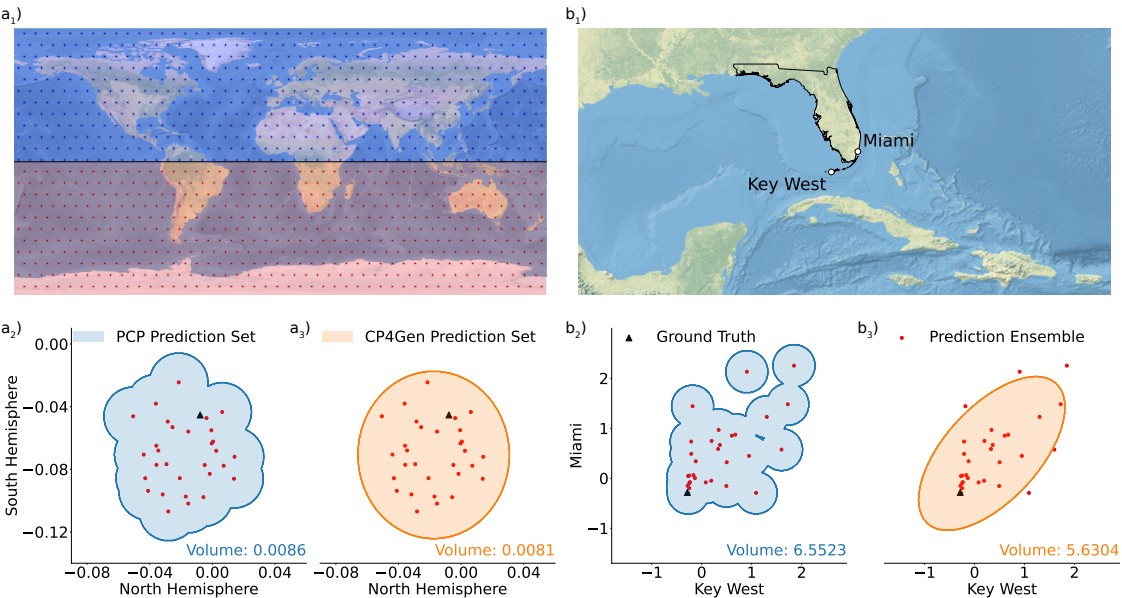

Figure 3: **Dimensionality Reduction and Prediction Set Demonstration on (a) Precip-2 and (b) Precip-2-N.** (a$_1$) Precip-2 is obtained by taking average of precipitation values over north (blue shaded) and south (red shaded) hemispheres. (b$_1$) Precip-2-N is by taking precipitation values at Miami and Key West. (a$_{2-3}$ and b$_{2-3}$) On both datasets, CP4Gen gives a smaller and simpler prediction set. The volume advantage is more evident when response two dimensions are correlated.

**High-dimensional Response: Climate Emulation** Lastly, we evaluate our method on a climate emulation task, built from climate simulations (Olonscheck et al., 2023, MPI) (c.f. Appendix B.1 for details). The goal is to model the response distribution of June daily mean precipitation given June monthly globally averaged temperature of that year. The response is a vector of precipitation values on a global grid of $1.865° \times 1.875°$ resolution, resulting in a response vector of size $96 \times 192$.

In this work, to ease the prediction set volume evaluation, dimensionality reduction is applied to the high-dimensional responses from the precipitation emulation task. Four different dimensionality reduction methods

are considered. `Precip-2` reduces the response to a 2-d vector by taking the mean of precipitation over the North and South Hemispheres separately. `Precip-2-N` (nearby) also reduces the response to a 2-d vector, but does so by taking precipitation values at two nearby cities: Miami and Key West. `Precip-3` returns a 3-d response vector by taking the mean of precipitation over the north, nouth and central bands of the globe separately. `Precip-4` reduces the response to a 4-d vector by taking the mean of precipitation over four quadrants separately. CP is then performed on each kind of the reduced-dimensional responses.

## 5 Results

CP4Gen's performance on all the datasets is summarized in Figure 2. Metric numerics are reported in Tables 1, 2, and 3.

**CP4Gen prediction sets are both lower complexity and lower volume.** In all the datasets, CP4Gen's empirical coverage rate is very close to the theoretical marginal coverage rate of $1 - \alpha = 0.9$ (c.f. Figure 2a). This verifies that CP4Gen's prediction set is valid. Given the similar coverage rate, CP4Gen's prediction set volume and structural complexity are generally lower than PCP's. In our experiments, the prediction ensemble size $M$ is set to 30 for all the datasets. An ablation study on $M$ is provided in Appendix E. Following Equation equation 5, PCP naturally gives structural complexity of 30 in all the experiments. On the other hand, even though the number of Gaussian components $K$ in CP4Gen is optimized for the set volume in each dataset, the resultant structural complexity is still significantly less than 30, as illustrated by Figure 2b on a normalized scale. One interesting result is that CP4Gen also recovers $K = 30$ in the `Taxi` dataset. demonstrates that CP4Gen can match PCP in terms of structural complexity when it is necessary to minimize volume. Since PCP is a special case of CP4Gen when $K = M$, unsurprisingly, PCP's prediction set volume is lower bounded by CP4Gen (Figure 2c). A visualization of the prediction sets for `S-Curve` and `25-Gaussians` is provided in Figure 4, highlighting that CP4Gen's prediction set is of simpler structure and lower volume compared to PCP.

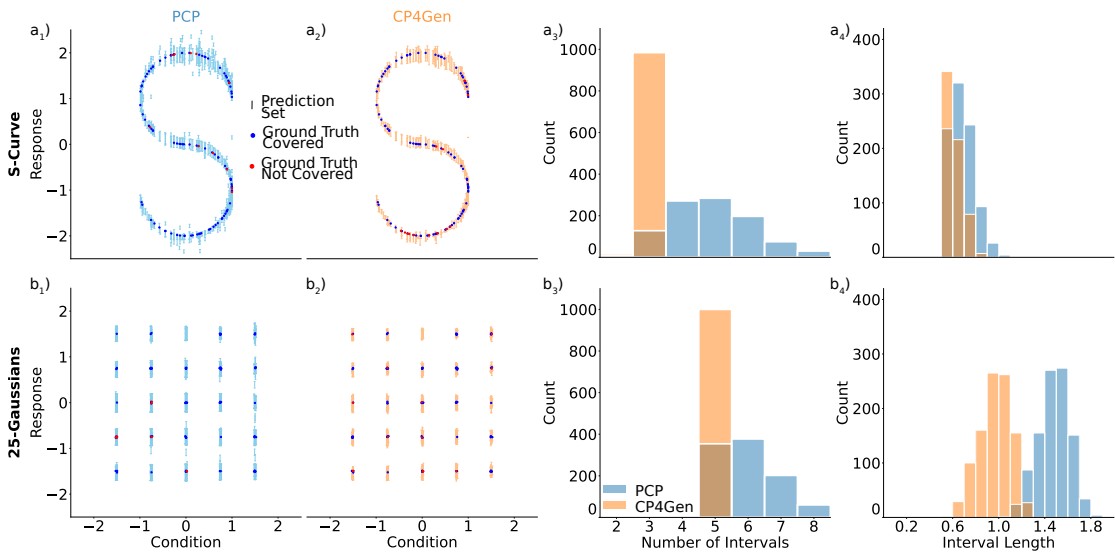

Figure 4: **Prediction set ($\alpha = 0.1$) visualization on synthetic datasets: S-Curve and 25-Gaussians.** Results for (a) S-Curve and (b) 25-Gaussians. Prediction sets by PCP and CP4Gen are visualized for 100 test data samples at the first two columns, ($a_1$, $b_1$) for PCP and ($a_2$, $b_2$) for CP4Gen. In addition, the last two columns show the histogram of the number of disjoint intervals constructing each prediction set ($a_3$, $b_3$) and the histogram of the volume of each final prediction set ($a_4$, $b_4$). Clearly, CP4Gen produces prediction sets with lower structural complexity (i.e., fewer disjoint intervals) and smaller volume than PCP.

**CP4Gen outperforms PCP on responses of various dimensionality.** In the precipitation emulation task, the global precipitation response is reduced to 2-d, 3-d, and 4-d responses to facilitate the evaluation.

CP4Gen outperforms PCP on all versions of the reduced responses, with a much lower prediction set volume and simpler set structure. In particular, volume reduction by CP4Gen gets more significant as the dimension increases. It implies that CP4Gen is able to consistently construct high-quality prediction sets regardless of the response dimensionality.

**CP4Gen reduces the volume further when the response dimensions are correlated.** Comparing the results on `Precip-2` and `Precip-2-N`, we observe that CP4Gen improves on PCP's prediction set volume further when the response has strongly correlated dimensions. As can be observed on the case study shown in Figure 3, for `Precip-2-N`, where the precipitation in Miami and Key West is strongly correlated, CP4Gen reduces the prediction set volume by 14% compared to PCP. On the contrary, for `Precip-2`, where north and south hemispheres precipitation is weakly correlated, the reduction is only 6%.

**A trade-off between structural complexity and volume.** As shown in our experiments (e.g., Tables 1), increasing $K$ from 1 improves the fidelity of the density estimate and typically reduces volume, but only up to the point where additional clusters begin to overfit to the sampling noise. When $K$ becomes too large, mixture components gradually collapse onto individual samples, covariance estimates shrink, and the resulting prediction set becomes a union of many small ellipsoids. This sharply increases structural complexity with diminishing returns in volume reduction. Therefore, too low or too high CP4Gen complexity would lead to a over-conservative prediction.

## 6 Discussion and Conclusion

We present a conformal prediction approach tailored to generative models with ensemble outputs. The procedure first fits a conditional distribution to the ensemble and then computes nonconformity scores from the fit. Building on this design, we derive a novel method termed CP4Gen, which uses a $K$-component Gaussian mixture estimator and defines a nonconformity score as the negative log-density from the nearest mixture component. This choice significantly mitigates the sensitivity to outlier samples and curbs the prediction set structural complexity, yielding sets that are sharper, more interpretable, and easier to optimize for downstream applications.

Empirically, CP4Gen produces prediction sets not only of simpler structure but also of smaller volume than PCP. This is due to a fundamental difference between CP4Gen and PCP in terms of ensemble density estimation, where CP4Gen has extra capacity (by varying $K$) to ensure better convergence to ensemble distributions when $M$ increases, whereas PCP (with a fixed $K$ to $M$) is likely to maintain a distribution bias (which translates into a volume bias, e.g. Figure 5). We analyze this phenomenon in a simplified setting and theoretically conclude that keeping $K$ properly small often improves sharpness and yields prediction sets with simpler geometry (c.f. Appendix F). Below, we discuss several remaining directions for future work.

**Density Estimation** While the Gaussian mixture in CP4Gen is able to capture multimodal structure of ensemble distributions, there are still many distributional structures untapped such as spatial localization and Markovian dependence. They could be utilized by alternative density estimation methods to improve density estimation and produce sharper conformal sets in high dimensions (Tipping & Bishop, 1999; Ghahramani & Hinton, 1996; Zargarbashi et al., 2023).

**Online Calibration** CP4Gen assumes calibration and test exchangeability, which is usually violated by real world data. CP4Gen can be extended to accommodate distribution shift by incorporating modern techniques from robust and online conformal prediction, enabling valid calibration in evolving generative regimes. These extensions would broaden the applicability of CP4Gen to streaming, sequential, and online prediction tasks (Zaffran et al., 2022; Xu & Xie, 2023; Gibbs & Candès, 2024; Aolaritei et al., 2025).

**Conditional Guarantees** When the conditional distribution $P(Y|X)$ varies with $X$, CP4Gen, like many conformal methods, guarantees only finite-sample marginal coverage but not exact or asymptotic conditional coverage. CP4Gen could be combined with remedies, such as localization and reweighting strategies (Guan, 2023; Tibshirani et al., 2019), for improved conditional coverage.

Overall, our conformal prediction approach, demonstrated with CP4Gen, represents a significant advance in the intersection of conformal prediction and generative modeling. It offers researchers and practitioners a powerful tool for calibrating uncertainty associated with generative models.

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

## Appendix

The appendix is organized as follows:

- In Appendix A, we provide an algorithmic description of CP4Gen.

- In Appendix B, we describe the datasets and generative model used in the experiments in greater details.

- In Appendix C, we present the original tabular data for CP method performance comparison in Figure 2.

- In Appendix D, we present an additional comparison between CP4Gen and HD-PCP.

- In Appendix E, we demonstrate an ablation study on the effect of generative ensemble size to CP method performance.

- In Appendix F, we provide some theoretical results developed in this work and their proofs.

# A   Algorithmic Description of CP4Gen

The pseudo-code of implementing CP4Gen in practice is presented in Algorithm 1.

---

**Algorithm 1:** CP4Gen: Conformal Prediction for Generative Models via $K$-means Clustering

---

**Input:** Dataset $\mathbf{D} = \{(X_i, Y_i)\}_{i=1}^{N}$, split into preliminary set $\mathbf{D}_p$ and calibration set $\mathbf{D}_c$; nominal level $\alpha$;
ensemble size $M$; number of clusters $K$; nugget $\beta^2 > 0$.
**Output:** Predictive set $\widehat{C}_\alpha(X^\star)$ for a test covariate $X^\star$.

**Step I: Fit Generative Model**
Train a conditional generative model $q(Y|X)$ on $\mathbf{D}_p$.

**Step II: Compute Nonconformity Scores**
    **for** $(X_i, Y_i) \in \boldsymbol{D}_c$ **do**

```
// Draw an ensemble and cluster it
```
Sample $\hat{\mathbf{Y}}_i = \{\hat{Y}_i^1, \ldots, \hat{Y}_i^M\} \sim q(Y \,|\, X_i)$;
Run $K$-means on $\hat{\mathbf{Y}}_i$;
Obtain $K$ clusters, each with weight $w_i^k$, means $\boldsymbol{\mu}_i^k$, and covariances $\boldsymbol{\Sigma}_i^k$;
```
// Negative log mixture density score (with a small nugget)
// Mixture density approximated by its dominant component for easier prediction
    set inversion
```
$$s_i \leftarrow -\log\left( \max_{1 \leq k \leq K} w_i^k \, \mathcal{N}\big(Y_i; \boldsymbol{\mu}_i^k, \boldsymbol{\Sigma}_i^k + \beta^2 I\big)\right).$$

Compute the $(1-\alpha)$ empirical quantile:

$$Q_{1-\alpha} \;\leftarrow\; \text{Quantile}_{1-\alpha}\big(\{s_i\}_{(X_i, Y_i) \in \mathbf{D}_c} \cup \{\infty\}\big).$$

**Step III: Predictive set at test time**
Sample $\hat{\mathbf{Y}}_\star = \{\hat{Y}_\star^1, \ldots, \hat{Y}_\star^M\} \sim q(Y \,|\, X^\star)$;
Run $K$-means on $\hat{\mathbf{Y}}_\star$ and obtain $\{w_\star^k, \boldsymbol{\mu}_\star^k, \boldsymbol{\Sigma}_\star^k\}_{k=1}^{K}$;
Define the score at $Y$:

$$s_\star(Y) \;=\; -\log\left( \max_{1 \leq k \leq K} w_\star^k \, \mathcal{N}\big(Y; \boldsymbol{\mu}_\star^k, \boldsymbol{\Sigma}_\star^k + \beta^2 I\big)\right);$$

Return the prediction set:

$$
\begin{aligned}
\widehat{C}_\alpha(X^\star) &= \big\{Y : s_\star(Y) \leq Q_{1-\alpha}\big\} \\
&= \bigcup_{k=1}^{K} \big\{Y : (Y - \boldsymbol{\mu}_\star^k)^\top (\boldsymbol{\Sigma}_\star^k + \beta^2 I)^{-1}(Y - \boldsymbol{\mu}_\star^k) \leq r_k\big\},
\end{aligned}
$$

where $r_k$ is the threshold implied by $Q_{1-\alpha}$ for component $k$.

---

*Remark* A.1. In Algorithm 1, we employ $K$-means to fit the Gaussian mixture model. It is worth noting that $K$-means relies on hard assignments, which can yield biased estimates of cluster means even with infinite data, particularly when the mixture components are not well separated (Jin & Malthouse, 2015). As an alternative, the expectation maximization (EM) algorithm is a widely used approach for density estimation; however, it typically requires a larger computational budget for convergence, and may still be trapped in a local minima of estimation (Jin et al., 2016). In experiments, we find that $K$-means produces clean and fast density fits across all tested datasets. An interesting future direction is to investigate more efficient yet accurate alternatives to $K$-means for mixture modeling.

# B Technical Details

## B.1 Datasets

Here we provide more details on the datasets used in the experiments.

### B.1.1 Synthetic Dataset

Synthetic datasets used in experiments are classic `25-Gaussians`, `8-Gaussians`, `Moon`, `Circle`, `Spiral`, and `S-Curve`. Each dataset has two dimensions: one is treated as the input and the other as the response. They are all generated from `scikit-learn` with the following parameters:

- `25-Gaussians`: 25 Gaussian clusters evenly spaced on a $5 \times 5$ grid, each with standard deviation $\sigma = 0.01$;

- `8-Gaussians`: 8 Gaussian clusters evenly spaced on a circle, each with standard deviation $\sigma = 0.01$;

- `Moon`: two moon-shaped clusters with Gaussian noise level $\sigma = 0.01$;

- `Circle`: two concentric circles with radius ratio $0.7 : 1$;

- `Spiral`: a spiral cluster as a slice of a 3-d swiss roll distribution;

- `S-Curve`: a S-curve cluster without Gaussian noise.

Each of them is generated with 5000 samples with 3000 for training, 1000 for calibration, and 1000 for testing.

### B.1.2 Real-World Dataset

Real-world data experiments are conducted on 1-d response public-domain datasets: bike sharing data (`Bike`), physicochemical properties of protein tertiary structure (`Bio`), blog feedback (`Blog`), and Facebook comment volume, variants one (`Facebook-1`) and two (`Facebook-2`), medical expenditure panel survey number 19 (`MEPS-19`), number 20 (`MEPS-20`), and number 21 (`MEPS-21`) (Romano et al., 2019), and temperature forecast data (`Temperature`). Each of them consists of multiple columns of features and one column of response.

- `Bike`: to predict the number of bike rentals given the weather and time(Fanaee-T, 2013).

- `Bio`: to predict the solubility of protein given physicochemical properties(Rana, 2013).

- `Blog`: to predict the number of comments given the blog features(Krisztian Buza, 2014).

- `Facebook-1`: to predict the number of comments given the facebook post features, variant one(Kamaljot Singh, 2015).

- `Facebook-2`:to predict the number of comments given the facebook post features, variant two.

- `MEPS-19`: to predict the medical expenditure given the patient features from panel survey number 19 (Agency for Healthcare Research and Quality, 2016).

- `MEPS-20`: to predict the medical expenditure given the patient features from panel survey number 20.

- `MEPS-21`: to predict the medical expenditure given the patient features from panel survey number 21.

- `Temperature`: to predict next day air temperature given numerical weather forecasts and in-situ observations(Cho et al., 2020).

Expriments are also conducted on 2-d response datasets: `Taxi` dataset and `Energy` dataset. Their detailed descriptions are already provided in the main paper (c.f. Section 4.3). All datasets are splitted by a ratio $6 : 2 : 2$ for training, calibration, and testing.

### B.1.3 Climate Emulation Dataset

The CP methods are also evaluated on a climate emulation task, which is built from a climate simulation dataset (Olonscheck et al., 2023). The goal is to model a response distribution of June daily mean precipitation given June monthly globally averaged temperature of that year. The response is a vector of precipitation values on a global grid of $1.865° \times 1.875°$ resolution, resulting in a response vector of size $96 \times 192$. The input is a scalar, which is the monthly and globally averaged temperature in June. The training data are from a climate simulation from 1950 to 2100. The calibration set is the data from 50 new simulations spanning between 1950 and 2014; and the rest from 2015 to 2100 is reserved as the test set.

### B.2 Conditional Generative Models

We used flow-matching (Lipman et al., 2024) to train a conditional generative model on the synthetic and real-world datasets. The conditional flow-matching generator is trained with an explicit Gaussian path. For each condition-target $(x, y)$ pair, we create intermediate states by mixing the ground-truth target with Gaussian noise according to a time schedule: $y_t = t \cdot y + \sqrt{1-t} \cdot \epsilon$, where $\epsilon$ is Gaussian standard noise. This mixing is used to produces a vector field $v$ to flow a sample from a standard Gaussian distribution to a desired response conditional on $x$. The vector field does not have a close form; it needs to be approximated ($v_\theta$), here with a simple MLP. This MLP consists of five linear layers with ReLU activation and is trained by randomly sampling $t \sim \text{Unif}[0, 1]$, with loss function

$$\mathcal{L} = \mathbb{E}_{t,\epsilon,y,x} \left[ \|v_\theta(y_t, t, x) - v^*(y_t, t)\|^2 \right], \tag{7}$$

and ground truth conditional flow $v^*(y_t, t) = \frac{d}{dt} y_t = y - \frac{\epsilon}{2\sqrt{1-t}}$. At inference time, we start from standard Gaussian noise and integrate over the learned vector field from $t = 0$ to $t = 1$ using forward Euler with a fixed step, yielding conditionally generated samples. These flow-matching models are trained and sampled on an HPC cluster with RTX 8000 NVIDA GPUs.

Note that the samples generated for the climate emulation task stem from another flow matching-model detailed in Wang et al. (2025).

## C CP Method Evaluation in Tabular Format

This section shows additional results for the CP methods on the previously described datasets. Their performance is evaluated in terms of marginal coverage, prediction set volume, and structural complexity. The detailed results are shown in Table 1, Table 2, and Table 3. They are also visualized in Figure 2 in the main paper.

Table 1: **CP Performance on Synthetic Datasets.** PCP and CP4Gen are evaluated on synthetic datasets: 25-Gaussians, 8-Gaussians, Moon, Circle, Spiral, and S-Curve in terms of prediction set structural complexity, empirical coverage, and volume. Both PCP and CP4Gn generally achieve desired coverage rate of 90% ($\alpha = 0.1$). CP4Gen outperforms PCP by giving prediction sets of smaller volume and less structural complexity.

| Metric | Method | 25-Gaussians | 8-Gaussians | Moon | Circle | Spiral | S-Curve |
|--------|--------|--------------|-------------|------|--------|--------|---------|
| coverage | PCP | 0.90 | 0.91 | 0.90 | 0.91 | 0.93 | 0.90 |
| coverage | CP4Gen | 0.88 | 0.90 | 0.90 | 0.87 | 0.91 | 0.87 |
| complexity | PCP | 30.0 | 30.0 | 30.0 | 30.0 | 30.0 | 30.0 |
| complexity | CP4Gen | 5.0 | 2.0 | 2.0 | 4.0 | 5.0 | 3.0 |
| volume | PCP | 1.48 | 0.41 | 0.25 | 0.40 | 6.45 | 0.66 |
| volume | CP4Gen | 0.97 | 0.36 | 0.21 | 0.37 | 5.56 | 0.54 |

Table 2: **CP Performance on Real World Datasets.** The performance of PCP and CP4Gen is assessed on real world datasets: MEPS-19, MEPS-20, MEPS-21, Facebook-1, Facebook-2, Bio, Blog, Temperature, Bike, Taxi, and Energy, focusing on prediction set structural complexity, empirical coverage, and volume. Both methods achieve the target coverage rate of 90% ($\alpha = 0.1$). CP4Gen produces prediction sets with reduced volume and lower structural complexity compared to PCP.

| Metric | Method | MEPS-19 | MEPS-20 | MEPS-21 | Facebook-1 | Facebook-2 |
|---|---|---|---|---|---|---|
| coverage | PCP | 0.90 | 0.90 | 0.91 | 0.91 | 0.90 |
| coverage | CP4Gen | 0.90 | 0.90 | 0.91 | 0.91 | 0.90 |
| complexity | PCP | 30.0 | 30.0 | 30.0 | 30.0 | 30.0 |
| complexity | CP4Gen | 1.0 | 1.0 | 1.0 | 1.0 | 1.0 |
| volume | PCP | 29.33 | 31.34 | 32.04 | 12.92 | 11.39 |
| volume | CP4Gen | 26.42 | 28.30 | 31.16 | 10.24 | 9.55 |

| Metric | Method | Bio | Blog | Temperature | Bike | Taxi | Energy |
|---|---|---|---|---|---|---|---|
| coverage | PCP | 0.90 | 0.91 | 0.90 | 0.90 | 0.90 | 0.95 |
| coverage | CP4Gen | 0.90 | 0.91 | 0.90 | 0.90 | 0.90 | 0.94 |
| complexity | PCP | 30.0 | 30.0 | 30.0 | 30.0 | 30.0 | 30.0 |
| complexity | CP4Gen | 2.0 | 1.0 | 1.0 | 1.0 | 30.0 | 1.0 |
| volume | PCP | 9.62 | 16.08 | 2.50 | 147.22 | 0.003 | 23.30 |
| volume | CP4Gen | 8.94 | 13.84 | 2.23 | 120.93 | 0.003 | 21.83 |

Table 3: **CP Performance on Climate Emulation.** The performance of PCP and CP4Gen is assessed on four types of dimensionality reduced precipitation responses: Precip-2, Precip-2-N, Precip-3, and Precip-4. Both methods typically achieve the target coverage rate of 90% ($\alpha = 0.1$). CP4Gen demonstrates superior performance compared to PCP in terms of reducing prediction set volume and structural complexity.

| Metric | Method | Precip-2 | Precip-2-N | Precip-3 | Precip-4 |
|---|---|---|---|---|---|
| coverage | PCP | 0.85 | 0.91 | 0.92 | 0.89 |
| coverage | CP4Gen | 0.84 | 0.91 | 0.92 | 0.88 |
| complexity | PCP | 30.0 | 30.0 | 30.0 | 30.0 |
| complexity | CP4Gen | 1.0 | 1.0 | 1.0 | 1.0 |
| volume | PCP | 0.0082 | 5.8275 | 0.0006 | 0.0003 |
| volume | CP4Gen | 0.0072 | 5.4204 | 0.0005 | 0.0002 |

# D    Comparison between CP4Gen and HD-PCP

This section shows an additional comparison between CP4Gen and HD-PCP on the previously described datasets. Their performance is evaluated in terms of marginal coverage, prediction set volume, and structural complexity. The detailed results are shown in Table 4.

HD-PCP (Wang et al., 2022) is an extension of PCP. It requires both generated samples and per-sample confidence scores. It first filters out the samples with the lowest confidence scores, and then applies PCP on the remaining samples. By dropping low-confidence samples, HD-PCP effectively reduces the fragmentation of

the resulting prediction set and improves the robustness to outliers. The additional access to confidence scores gives HD-PCP an unfair advantage over CP4Gen. However, it also makes HD-PCP less generally applicable than CP4Gen as sample confidence scores are usually not available for many generative or simulation-based ensemble models.

In Table 4, we observe that even though CP4Gen is at a disadvantage due to the lack of confidence scores, it still achieves comparable performance to HD-PCP in terms of prediction set volume given the same coverage rate. These results show that both the filtering in HD-PCP and the clustering in CP4Gen effectively alleviate the issue of outliers and make the algorithms more robust, leading to improved sharpness. However, we emphasize that HD-PCP still has a much higher structural complexity than CP4Gen. This comparison provides strong evidence that the improved performance of CP4Gen is not purely due to outlier cleaning but a better distribution modeling choice made possible by GMM and $K$-means. We also ran experiments with CP4Gen applied to filtered ensemble samples; in these experiments, we still see improvement over HD-PCP. It further verifies that complexity/volume reductions are the result of these two orthogonal ingredients: a better modeling choice and a cleaner sample set.

Table 4: **CP4Gen vs. HD-PCP Performance Comparison** CP4Gen and HD-PCP are compared on a comprehensive set of datasets, including synthetic and real-world ones, in terms of prediction set structural complexity, empirical coverage, and volume. Both CP4Gen and HD-PCP generally achieve desired coverage rate of 90% ($\alpha = 0.1$); and give prediction sets of comparable volumes. However, CP4Gen outperforms HD-PCP by creating prediction sets of much less structural complexity.

| Dataset | CP4Gen Cov. | HD-PCP Cov. | CP4Gen Vol. | HD-PCP Vol. | CP4Gen $K$ | HD-PCP $K$ |
|---|---|---|---|---|---|---|
| 25-Gaussians | 0.91 | 0.91 | 1.08 | 1.49 | 5 | 28 |
| 8-Gaussians | 0.91 | 0.89 | 0.37 | 0.41 | 2 | 30 |
| Moon | 0.90 | 0.91 | 0.22 | 0.21 | 2 | 15 |
| Circle | 0.90 | 0.89 | 0.40 | 0.37 | 30 | 28 |
| Spiral | 0.90 | 0.90 | 5.24 | 5.49 | 5 | 28 |
| S-Curve | 0.90 | 0.89 | 0.59 | 0.59 | 3 | 28 |
| Meps-19 | 0.90 | 0.90 | 26.42 | 26.36 | 1 | 18 |
| Meps-20 | 0.90 | 0.90 | 28.30 | 26.64 | 1 | 15 |
| Meps-21 | 0.91 | 0.91 | 31.16 | 27.65 | 1 | 18 |
| Facebook-1 | 0.91 | 0.91 | 10.24 | 11.24 | 1 | 28 |
| Facebook-2 | 0.90 | 0.90 | 9.55 | 9.41 | 1 | 18 |
| Bio | 0.90 | 0.90 | 8.94 | 8.77 | 2 | 28 |
| Blog | 0.91 | 0.91 | 13.84 | 13.92 | 1 | 18 |
| Temperature | 0.90 | 0.92 | 2.23 | 2.16 | 1 | 15 |
| Bike | 0.90 | 0.90 | 120.93 | 113.76 | 1 | 18 |
| Energy | 0.94 | 0.90 | 21.83 | 11.01 | 1 | 9 |

# E    Ablation Study on Ensemble Size

In the main paper, the prediction ensemble size $M$ is set to 30 for all experiments. To study the effect of $M$, we ablate $M$ on two selected datasets: `25-Gaussians` and `Bio`. The results are shown in Figure 5. We observe that prediction set coverage stays close to the theoretical marginal coverage rate of $1 - \alpha = 0.9$ for all $M$, indicating that PCP and CP4Gen are valid conformal prediction methods. We also observe that both PCP and CP4Gen's prediction set volume decrease as $M$ increases when $M$ is small; and PCP's prediction set volume is lower bounded by CP4Gen's. It is expected since PCP is a special case of CP4Gen when $K = M$ and $K$ is optimized for optimal volume within CP4Gen. Interestingly, when $M$ is greater than 20, CP4Gen's prediction set volume continues to decrease while PCP's converges to a constant. It is consistent with our analysis in the main paper that PCP induces a response distribution estimation bias which prevents it from generating prediction sets asymptotically sharper as $M$ increases. In terms of prediction set complexity,

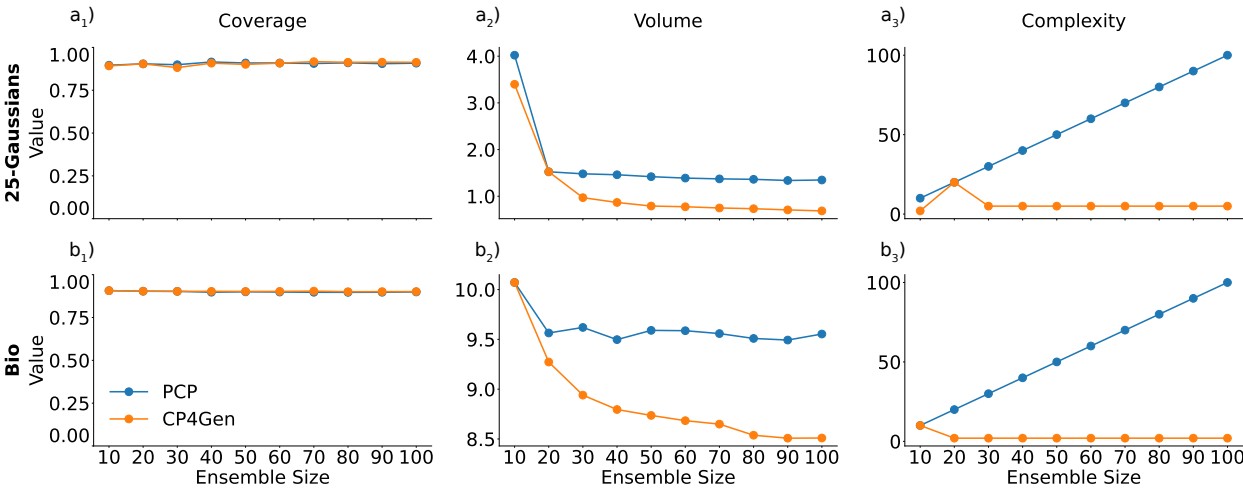

Figure 5: **A study on the effect of prediction ensemble size.** Results for (a) 25-Gaussians and (b) Bio. From left to right, prediction set coverage rate ($a_1$, $b_1$), volume ($a_2$, $b_2$), and structural complexity ($a_3$, $b_3$) are plotted separately each as a function of prediction ensemble size. The prediction ensemble size is varied on a grid of $[10, 20, 30, \ldots, 100]$. Both CP methods' coverage rate stays flat close to 90% ($\alpha = 0.1$) for all ensemble sizes . As ensemble size increases, both methods' prediction set volume decreases. Interestingly, CP4Gen's set complexity converges to a constant (5 for 25-Gaussians and 2 for Bio) when ensemble size is larger than 20, while PCP's complexity scales linearly with ensemble size.

PCP's scales linearly with $M$ as decided by its score function formulation. In contrast, CP4Gen's complexity converges to a constant when $M$ is greater than 20. It may be because a large ensemble size is required for $K$-means to find the correct number of modes $K$ in the underlying response distribution, which is necessary for CP4Gen to give an accurate estimation of the distribution, and thus generate prediction sets with the optimal volume.

## F  Theoretical Results and Proofs

By the construction of the predictive set defined in Equation (4), CP4Gen has a guaranteed marginal coverage as shown in the following Theorem.

**Theorem F.1.** *Suppose the calibration data* $(X_i, Y_i, \hat{\mathbf{Y}}_i)$, $i \in \{1, 2, \ldots, N\}$ *and the testing data* $(X, Y, \hat{\mathbf{Y}})$ *are exchangeable, then the predictive set defined in Equation* (4) *satisfies*

$$P_{X,Y,\hat{\mathbf{Y}}}(Y \in \hat{C}_\alpha(X)) \geq 1 - \alpha. \tag{8}$$

*When the scores are distinct almost surely,*

$$P_{X,Y,\hat{\mathbf{Y}}}(Y \in \hat{C}_\alpha(X)) \leq 1 - \alpha + \frac{1}{N+1}. \tag{9}$$

Theorem F.1 shows that the marginal coverage of the prediction set is sharp up to $\mathcal{O}(\frac{1}{N+1})$ for distinct scores, which is true when the conditional distribution $P_{Y|X}$ is absolutely continuous. The theorem follows from the following lemma, and the proofs of both results are provided in Romano et al. (2019) and Tibshirani et al. (2019).

**Lemma F.2.** *Suppose* $Z_1, \ldots, Z_n, Z_{n+1}$ *are scalar random variables that are exchangeable and almost surely distinct, then for* $\beta \in [0, 1]$,

$$\beta \leq P(Z_{n+1} \leq Q_\beta(Z_{1:n} \cup \{\infty\})) \leq \beta + \frac{1}{n+1}. \tag{10}$$

**Lemma F.3.** *Let $f$ be a probability density on $\mathbb{R}^d$, let $\lambda(\cdot)$ denote the Lebesgue measure, and let*

$$C_\alpha^{\text{HD}} := \{y : f(y) \geq \tau_\alpha\} \quad \text{with} \quad \int_{C_\alpha^{\text{HD}}} f(y)dy = 1 - \alpha \tag{11}$$

*be the highest-density (HD) set corresponding to probability $1-\alpha$. For simplicity, we assume the highest-density set $C_\alpha^{HD}$ for chosen $f$ and $\alpha$ exists and is unique up to sets of measure zero. Then the following statements hold:*

1. ***Deterministic set.** For any measurable $C \subset \mathbb{R}^d$ with $\int_C f \geq 1 - \alpha$,*

$$\lambda(C) \geq \lambda(C_\alpha^{\text{HD}}), \tag{12}$$

   *with equality iff $C = C_\alpha^{\text{HD}}$ (up to a null set).*

2. ***Random set.** If $C$ is a random measurable set such that $\mathbb{E}_C \int_C f \geq 1 - \alpha$, then*

$$\mathbb{E}_C \lambda(C) \geq \lambda(C_\alpha^{\text{HD}}), \tag{13}$$

   *with equality iff $C = C_\alpha^{\text{HD}}$ almost surely (up to a null set).*

*Proof.* (1) The statement follows from the fact that any HD set minimizes Lebesgue measure among all measurable sets with at least $1 - \alpha$ probability mass under $f$. (2) Define the selection function $g(y) := P(y \in C) \in [0, 1]$. Then

$$\mathbb{E}_C \lambda(C) = \int g(y)dy,$$
$$\mathbb{E}_C \left[ \int_C f \right] = \int g(y)f(y)dy \geq 1 - \alpha. \tag{14}$$

Thus we are minimizing $\int g \, d\lambda$ over $g$ subject to $0 \leq g \leq 1$ and $\int gf \geq 1 - \alpha$. Form the Lagrangian with a scalar multiplier $\lambda \geq 0$ for the $1 - \alpha$ mass constraint and function multipliers $u(y), v(y) \geq 0$ for the pointwise bounds:

$$\mathcal{L}(g, \lambda, u, v) = \int [g - \lambda fg - ug + v(g - 1)]d\lambda + \lambda(1 - \alpha). \tag{15}$$

First-order optimality condition holds pointwise a.e.:

$$\frac{\delta\mathcal{L}}{\delta g}(y) = 1 - \lambda f(y) - u(y) + v(y) = 0. \tag{16}$$

And we also have the complementary slackness

$$u(y)g(y) = 0,$$
$$v(y)(g(y) - 1) = 0, \tag{17}$$
$$\lambda\left(\int fg - (1 - \alpha)\right) = 0.$$

From these, if $\lambda f(y) > 1$, then the first-order optimality condition plus non-negativity of $u(y), v(y)$ forces $v(y) > 0$, and the complementary slackness implies that $g(y) = 1$. Similarly, if $\lambda f(y) < 1$, the same reasoning implies that $g(y) = 0$; if $\lambda f(y) = 1$, then $v(y) = u(y) = 0$, and

$$1 - \lambda f(y) = 0 \quad \Rightarrow \quad f(y) = \tau := 1/\lambda. \tag{18}$$

As a result, minimizer $g^*$ must satisfy

$$\int fg^* = 1 - \alpha, \tag{19}$$

$$g^*(y) = \begin{cases} 1, & f(y) > \tau, \\ 0, & f(y) < \tau, \\ \theta(y), & f(y) = \tau, \ 0 \leq \theta(y) \leq 1. \end{cases} \tag{20}$$

Because we assume the highest-density set is unique, $\{y : f(y) = \tau\}$ must have measure 0. Thus the unique minimizer $g^*$ is given by the indicator of the $1 - \alpha$ HD set $g^* = \mathbb{I}_{C_\alpha^{\mathrm{HD}}}$. Therefore,

$$\mathbb{E}[\lambda(C)] \geq \int g^* d\lambda = \lambda(C_\alpha^{\mathrm{HD}}), \tag{21}$$

with equality iff $g = g^*$ almost everywhere, i.e. $C = C_\alpha^{\mathrm{HD}}$ almost surely (up to a null set). $\qquad\square$

The next proposition is introduced under simplified assumptions for exposition, but the underlying intuition applies broadly. Despite employing a few approximations, the proof elucidates the fundamental mechanism of the result.

**Proposition F.4.** *Assume the true conditional distribution $P(Y|X)$ is Gaussian with mean $\mu$ and covariance matrix $\Sigma$, and the generative model outputs i.i.d. ensemble data from $P(Y|X)$, then the conformal set given by CP4Gen gets asymptotically sharper as $M$ increases, while the set given by PCP does not.*

**Intuition.** Assume the generative model is correctly specified and the true conditional density $f$ is an input-ignorant Gaussian with parameters $\theta^\star = (\mu^\star, \Sigma^\star)$. In this case, PCP amounts to fitting the ensemble $\{\hat{Y}^m\}_{m=1}^M$ using a kernel density estimator (KDE) with a vanishing bandwidth $\sigma$, and using the negative log-density as the score function. By contrast, CP4Gen is equivalent to first fitting a Gaussian model to the ensemble and applying the same score function. The proof shows that the KDE fit is an inconsistent density estimate for all $M$ under the vanishing $\sigma$ assumption. On the other hand, the Gaussian fit, being correctly specified, provides a consistent (asymptotically unbiased) plug-in density estimate. Consequently, its conformal set converges to the true HD set of $f$ and is asymptotically sharp. Similar analysis can be extended to Mixture of Gaussian and general non-Gaussian target with careful justification.

*Proof.* We begin by observing that PCP and CP4Gen essentially differ in the density models used to fit the ensemble. For true label $Y$, consider the KDE log-sum and log-max scores from Equation (5):

$$-\log \sum_{m=1}^M \frac{1}{M} \mathcal{N}(Y; \hat{Y}^m, \sigma^2 I),$$
$$-\log \max_{1 \leq m \leq M} \frac{1}{M} \mathcal{N}(Y; \hat{Y}^m, \sigma^2 I). \tag{22}$$

PCP employs the minimum-distance score $\min_{1 \leq m \leq M} \|Y - \hat{Y}^m\|$, which is equivalent in ranking to the KDE log-max score, since conformal sets depend only on order statistics. Directly analyzing either the KDE log-max score or the minimum-distance score is challenging due to the non-smooth maximum operator. However, by choosing $\sigma$ appropriately for fixed $M$, one can control the gap between the KDE log-sum and log-max scores. Defining $m^*(y) = \arg\min_m \|y - \hat{Y}^m\|$, we obtain

$$\Delta(y) = \min_{m \neq m^*(y)} \frac{\|y - \hat{Y}^m\|^2 - \|y - \hat{Y}^{m^*}\|^2}{2\sigma^2},$$
$$0 \leq \log \sum_{m=1}^M \frac{1}{M} \mathcal{N}(y; \hat{Y}^m, \sigma^2 I) - \log \max_m \frac{1}{M} \mathcal{N}(y; \hat{Y}^m, \sigma^2 I) \leq \log(1 + (M - 1)e^{-\Delta(y)}). \tag{23}$$

Since $\{\hat{Y}^m\}_{m=1}^M$ are random samples from a continuous Gaussian distribution with fixed covariance, the distance between the two nearest neighbors scales as $\min_{m \neq n} \|\hat{Y}^m - \hat{Y}^n\| \asymp M^{-1/d}$. Thus we require $e^{-\Delta(y)} \to 0$, i.e. $\sigma = o(M^{-1/d})$, so that all non-nearest terms become exponentially negligible (Biau & Devroye, 2015). With such choice of $\sigma$, the approximation gap between the KDE log-sum and KDE log-max scores is uniformly controlled with high probability. This reduction allows us to focus on the log-sum score, whose concentration properties are more tractable.

On the other hand, consider the GMM log-sum and log-max scores in Equation (6):

$$-\log \sum_{k=1}^{K} w_k \, \mathcal{N}(Y; \boldsymbol{\mu}_k, \boldsymbol{\Sigma}_k),$$

$$-\log \max_{1 \leq k \leq K} w_k \, \mathcal{N}(Y; \boldsymbol{\mu}_k, \boldsymbol{\Sigma}_k). \tag{24}$$

Under the Gaussian target assumption, these two scores above coincide, and we may analyze the GMM log-sum score directly without adjustment. Thus, up to approximation and rank equivalence, both PCP and CP4Gen use the negative log-density as the conformal score function, but they fit the conditional distribution with different models.

We next show that PCP's estimated conditional law does not converge to the true Gaussian distribution, whereas CP4Gen's estimate does. In conformal prediction, the $n$-th calibration datum consists of $M$ i.i.d. draws from the Gaussian distribution. PCP fits these samples with a KDE $\hat{f}_n^{\mathrm{KDE}} = \sum_{m=1}^{M} \frac{1}{M} \mathcal{N}(\,\cdot\,; \hat{Y}^m, \sigma^2 I)$ using a vanishing bandwidth $\sigma = o(M^{-1/d})$.

For consistency of the kernel density estimator, the bandwidth must satisfy $\sigma \to 0$ and $M\sigma^d \to \infty$ (Chen, 2017). This rate is asymptotically slower than the regime required earlier for bounding the difference between the log-sum and log-max mixture scores. Consequently, for any $M$, one can always choose a bandwidth $\sigma$ small enough that the KDE evaluation fails to converge to the true Gaussian density.

By contrast, CP4Gen assumes a parametric form. It estimates the plug-in parameters $\hat{\theta}$ from the ensemble and fits a Gaussian model $\hat{f}_n^{\mathrm{GMM}} = \mathcal{N}(\,\cdot\,; \hat{\mu}, \hat{\Sigma})$. As $M$ increases, standard parametric theory yields

$$\sqrt{M}\,(\hat{\theta} - \theta^\star) \xrightarrow{d} \mathcal{N}\big(0,\, I(\theta^\star)^{-1}\big), \tag{25}$$

where $I(\theta^\star)$ is the Fisher information of the Gaussian parameter at $\theta^\star$. By the delta method, for each fixed $y$,

$$\sqrt{M}\,\big(\hat{f}_n^{\mathrm{GMM}}(y) - f^\star(y)\big) \xrightarrow{d} \mathcal{N}\big(0,\, \nabla g_y(\theta^\star)^\top I(\theta^\star)^{-1} \nabla g_y(\theta^\star)\big), \tag{26}$$

where $g_y(\theta) := f_\theta(y) = \mathcal{N}(y; \mu, \Sigma)$ is the density function seen as a function of $\theta$. Thus, $\hat{f}_n^{\mathrm{GMM}}$ converges asymptotically to the true Gaussian distribution with $M \to \infty$.

Finally, we connect conditional density estimation to the expected volume of conformal sets. Write $f^\star(\cdot)$ for the true conditional density and $\hat{f}(\cdot) = \hat{f}_{N+1}^\bullet(\cdot)$ for a plug-in estimator ($\bullet \in \{\mathrm{KDE}, \mathrm{GMM}\}$). When the number of calibration samples $N$ goes to infinity, the conformal threshold converges to the population threshold, so the conformal set can be written as the super-level set by Neyman-Pearson Lemma:

$$\hat{C}_\alpha = \{y : \hat{f}(y) \geq \tau\}, \tag{27}$$

where $\tau$ solves

$$\mathbb{E}_{\hat{f}} \int \mathbb{I}\{\hat{f}(y) \geq \tau\} f^\star(y) dy = 1 - \alpha. \tag{28}$$

Now for $\hat{f}(\cdot) = \hat{f}_{N+1}^{\mathrm{KDE}}(\cdot)$, since the KDE bandwidth shrinks at a fast speed $\sigma = o(M^{-1/d})$, the KDE estimator is inconsistent and does not converge to $f^\star$ for any $y$. Then, the highest-density (HD) sets of $\hat{f}$ and $f^\star$ differ on a set of nonzero Lebesgue measure. By Lemma F.3, among all deterministic measurable $A \subset \mathbb{R}^d$ with $\int_A f^\star = 1 - \alpha$, the minimal-volume set is the highest-density (HD) set $C_\alpha^{\mathrm{HD}}$, and for randomized $A$, $\mathbb{E}\lambda(A) \geq \lambda(C_\alpha^{\mathrm{HD}})$ with equality iff $A$ agrees with $C_\alpha^{\mathrm{HD}}$ up to a mull set almost surely. However, from Equation (27) we find that although $\hat{C}_\alpha$ attains the target mass $1 - \alpha$ under $f^\star$ on average, it cannot be the HD set of $f$ almost surely because its boundary is a level set of $\hat{f}$ rather than $f^\star$. Hence $\lambda(\hat{C}_\alpha) > \lambda(C_\alpha^{\mathrm{HD}})$, making PCP suboptimal.

In contrast, CP4Gen produces an asymptotically unbiased density estimator when the true law is Gaussian. Its conformal sets therefore converge to the highest-density set of $f^\star$, achieving asymptotically optimal volume. $\qquad\square$

While the analysis above focuses on large $M$ and input-ignorant Gaussian, the empirical results in Figure 5 demonstrate that CP4Gen yields tighter prediction sets for multi-modal targets with relatively small ensemble sizes ($M \geq 10$). This can be justified as follows: the reasoning in the proof naturally extends to mixtures of well-separated Gaussians. For more general distributions, both methods incur some bias; however, CP4Gen can flexibly mitigate this bias by increasing the number of mixture components $K$, leading to smaller conformal sets. Empirical evidence in Figure 5 support this analysis: as $M$ increases, PCP consistently produces larger prediction sets, whereas CP4Gen leverages structural information to generate sharper sets, even when the target distribution is not a Gaussian mixture.

