# OpenReview forum: "Conformal Prediction for Generative Models via Adaptive Cluster-Based Density Estimation"
_TMLR — Decision pending for TMLR_

### Review · Reviewer_2Zbe · 2026-05-04

**Summary Of Contributions:**

The paper proposes a conformal prediction method for conditional generative models that lack tractable densities. The core idea is straightforward: instead of treating each generated ensemble sample as its own ball-centered prediction region, it first clusters the ensemble by K-means, fits a Gaussian Mixture Model to those clusters, and defines nonconformity scores as the negative log-density of the dominant mixture component. The paper also introduces structural complexity as a new evaluation metric.

Strengthes:
1. Reframing PCP as a degenerate KDE and CP4Gen as a better-specified GMM is conceptually clean and gives the paper theoretical grounding beyond heuristic motivation.
2. Using structural complexity as a metric is interesting. This is a useful contribution. A fragmented prediction set is operationally harder to use than a simpler one.

Weaknesses:
1. K is chosen by minimizing prediction set volume on the preliminary set, but it seems that there is no theoretical guidance on how to set K, what happens when K is misspecified, or how sensitive results are to the grid search. More quantitative discussion would be helpful
2. It seems that coverage may fall below 1−α in the numerical results. More discussion would be beneficial.

**Audience:**

Yes

**Audience Explanation:**

The CP community is active and growing within TMLR's audience. This paper makes a concrete, testable contribution to an underexplored aspect: conformal prediction for implicit generative models. Researchers interested in diffusion model may be also inspired by this work.

**Claims And Evidence:**

Yes

**Claims Explanation:**

Overall, the evidence is largely convincing. Most claims such as CP4Gen consistently reducing structural complexity compared to PCP, are well supported. For instance, across every dataset in Tables 1–3, CP4Gen’s structural complexity is dramatically lower than PCP’s fixed complexity. However, while CP4Gen does achieve lower volume in most cases, the margins are often modest. This suggests that the evidence may not fully support the claim that CP4Gen consistently outperforms PCP in terms of prediction set volume.

**Requested Changes:**

See above comments

---

> ### Author Response · Authors · 2026-06-19
>
> We thank the reviewer for the constructive assessment. We agree that these two points should be further addressed: how the number of mixture components $K$ is chosen, and how to interpret empirical coverage that falls slightly below the nominal level in some experiments.
>
> **Choice and sensitivity of $K$.** We agree that more quantitative discussion of the role of $K$ is helpful. In the revision, we will add a sensitivity analysis that sweeps $K$ from 1 to the ensemble size $M$. The analysis will be provided as a supplementary figure in the appendix. The curve shows a clear v-shape: when $K$ is too small, distinct modes are merged, which enlarges the prediction set by including low-density regions between modes; when $K$ approaches $M$, the method approaches PCP, leading to more fragmented sets, which increases the prediction set volume again by including low-density regions supported by outliers. The sharpest sets are typically obtained at intermediate values of $K$ that match the geometry of the generated ensemble, i.e., the number of visually or statistically meaningful clusters.
>
> In addition, we want to reiterate that $K$ is a score-design hyper-parameter. It is your prior assumption on the structure of the ensemble distribution. It directly decides the resultant prediction sets' structure complexity and volume. You can choose $K$ based on your preference on complexity and volume. If you prefer a simple prediction set, you can choose $K$ as small as 1;
> if you prefer a sharp prediction set, you can choose $K$ by following the procedure used in our paper. But all choice of $K$ would lead to valid coverage.
>
> **Coverage below $1-\alpha$.** The theoretical guarantee with the conformal quantile defined in Eq. (4) is a marginal finite-sample statement over the randomness of the calibration/test samples under exchangeability. Therefore, the empirical coverage measured on one realized calibration/test split is itself random and can fall slightly below the nominal level. This is particularly relevant in our experiments, which — as is standard in conformal prediction literatures — report results on one fixed train/calibration/test split rather than averaging over many re-splits. We will revise the discussion to make this interpretation explicit.

---

### Review · Reviewer_6t9Z · 2026-05-10

**Summary Of Contributions:**

This paper proposes CP4Gen, a conformal prediction method tailored to conditional generative models that produce sample ensembles.

Strengths:
- The density-estimation view is illuminating and naturally subsumes probabilistic conformal prediction (PCP) as a special case.
- Extensive experiments on synthetic data, nine real-world tabular benchmarks, 2D-response tasks, and a climate emulation task.

Weaknesses:
- Proposition 3.1 is proved only for the Gaussian case. The paper claims extension to well-separated Gaussian mixtures and general non-Gaussian targets, but these are not formalized.
- The authors note that K-means yields biased cluster means under poor separation (Remark A.1) and that EM has similar performance with more cost, but no alternative is explored.

**Audience:**

Yes

**Audience Explanation:**

This work studies conformal prediction for generative models, which is in the scope of TMLR.

**Claims And Evidence:**

Yes

**Claims Explanation:**

The paper's central claims are well-supported by a combination of theoretical analysis and extensive empirical evidence.

**Requested Changes:**

- Formalize the theoretical claims beyond the Gaussian case.
- Explore alternatives to K-means, and report the performance of EM.

---

> ### Author Response · Authors · 2026-06-19
>
> We thank the reviewer for the positive and constructive assessment.
> We think both requested changes are reasonable and we will address them in the revision.
> In the revision, we will make it clearer which statements are theoretically justified and which are empirically justified,
> and we will have an explicit comparison between K-means and EM for fitting the Gaussian mixture used in CP4Gen.
>
> **Regarding theory beyond the Gaussian case.**
> We will revise Appendix F to state the claims more carefully.
> The finite-sample conformal coverage guarantee remains fully distribution-free under exchangeability.
> The asymptotic sharpness result in Proposition 3.1 is a formal result only for the single-mode Gaussian case.
>
> For multimodal distributions, the analysis is more subtle.
> The exact highest-density region of a mixture is a level set of the full log-sum density,
> whereas CP4Gen uses a dominant-component approximation to obtain a tractable prediction set with an explicit union-of-ellipsoids representation.
> PCP is also not asymptotically optimal in this setting, since its geometry remains restricted to a union of sample-centered balls.
> Thus, beyond the single-mode Gaussian case, we do not claim asymptotic optimality for either method.
> Instead, our claim is that CP4Gen better exploits the fitted Gaussian-mixture structure by tuning $K$,
> allowing it to approximate high-density regions more efficiently than PCP’s sample-centered construction.
> We support this claim empirically in Figure 5: across two datasets, CP4Gen yields prediction sets with smaller volume than PCP as the ensemble size increases.
> These results suggest that the efficiency gains captured by the theoretical unimodal Gaussian analysis also appear empirically in settings beyond the unimodal case.
>
> **Regarding alternatives to K-means.**
> We agree that the original manuscript mentioned EM without enough evidence.
> We will add an appendix ablation comparing K-means and EM as the mixture-fitting step, using the same generated ensembles.
> See the results attached below as well.
> The results show that EM gives similar coverage but generally bigger prediction-set volume, while requiring much more computation.
> We therefore keep K-means as the default because it is faster, simpler, and gives better prediction sets in our experiments.
> We will also revise Remark A.1 so that the discussion of EM is supported by the reported ablation rather than appearing as an unsupported statement.
>
> | Metric | Method | 25-Gaussians | 8-Gaussians | Moon | Circle | Spiral | S-Curve |
> |---|---|---|---|---|---|---|---|
> | coverage | EM             | 0.89 | 0.89 | 0.91 | 0.90 | 0.91 | 0.92 |
> | coverage | K-means    | 0.88 | 0.90 | 0.90 | 0.87 | 0.91 | 0.87 |
> | complexity | EM           | 5.0 | 3.5 | 3.0 | 4.3 | 4.0 | 4.0 |
> | complexity | K-means  | 5.0   | 2.0 | 2.0 | 4.0 | 5.0 | 3.0 |
> | volume | EM                 | 1.03 | 0.48 | 0.31 | 0.45 | 5.17 | 0.78 |
> | volume | K-means        | 0.97 | 0.36 | 0.21 | 0.37 | 5.56 | 0.54 |

---

### Review · Reviewer_ubQL · 2026-05-26

**Summary Of Contributions:**

This paper addresses calibrated uncertainty quantification for conditional generative models. Previous conformal prediction methods for this task suffer from overcoverage and are known to be sensitive to outliers. To overcome these limitations, the paper introduces a new conformal prediction method.　The key insight of this work is to reinterpret existing conformal prediction methods from the perspective of density estimation. In particular, the authors show that previous approaches can be viewed as density estimation using isotropic Gaussian distributions, which naturally enables an extension to Gaussian mixture models.　The proposed method is demonstrated on both synthetic datasets and high-dimensional climate emulators where it achieves improved prediction set quality.

**Audience:**

Yes

**Audience Explanation:**

Calibration of deep generative models is definitely of strong interest to the TMLR audience.

**Claims And Evidence:**

Yes

**Claims Explanation:**

Reinterpreting existing conformal prediction methods from the perspective of density estimation is an insightful contribution. This viewpoint can potentially inspire many essential extensions of existing methods and suggests a valuable future direction for conformal prediction research.

Demonstrating the effectiveness of the proposed method on high-dimensional climate emulation tasks is impactful and practically important.

Overall, the claims and arguments presented in this paper are well-motivated and coherent, and the work provides a promising new direction for the calibration of deep generative models.

**Requested Changes:**

Major

The discussion on extending the method to other density estimation approaches, particularly in high-dimensional settings, should be expanded further. In many practical applications, generative models are used to produce intrinsically high-dimensional outputs. The density-estimation perspective proposed in this work suggests the potential to address such challenges using alternative high-dimensional distributional metrics or estimators, such as MMD- or Wasserstein-based methods. Ideally, I would have liked to see empirical validation using density estimation methods beyond Gaussian mixture models. However, given that this is a TMLR submission, I do not consider such additional experiments strictly necessary for acceptance. Nevertheless, the paper should more clearly acknowledge that the current experiments are limited to Gaussian mixture–based density estimation and emphasize that one of the main contributions of this work is to highlight an important future research direction toward more expressive density estimation approaches for conformal prediction.

Minor

The approximate connection between Equations (5b) and (6) is explained intuitively in the paper, but the argument would be clearer if it were also supported by a more explicit mathematical derivation.

---

> ### Author Response · Authors · 2026-06-18
>
> We thank the reviewer for the thoughtful and encouraging assessment.
> We agree that the density-estimation perspective extends beyond the estimator we employ in our experiments.
> In the revision, we will make it clearer that our experiments are limited to Gaussian-mixture-based density estimation,
> and highlight that extending to more expressive density models is a promising direction for future work.
>
> **Regarding the scope of density estimation methods.**
> We agree that the current empirical instantiation is limited to Gaussian/Gaussian-mixture-based density estimation.
> In the revised manuscript, we will expand the Discussion section to make this limitation explicit.
> Additionally, we will emphasize that CP4Gen should be viewed as one concrete instantiation of the broader sample-based density-estimation framework,
> rather than as an exhaustive treatment of all possible density estimators.
>
> We will also have a short discussion of possible alternatives,
> including structured Gaussian-mixture variants and MMD- or Wasserstein-based scores, especially in high dimension.
> The main point we will make is that prediction set validity itself is not the obstacle: once a score is fixed, split conformal calibration gives marginal coverage under exchangeability.
> MMD and Wasserstein-based scores can be cheaper to compute than fitting and evaluating more elaborate density models.
> However, they will introduce additional hyper-parameters, such as kernel function and bandwidths.
> We also note that a direct Wasserstein score $W_p(\delta_y,\hat q_x)$ is less suitable for multimodal outputs because it averages the distance from $y$ to all generated samples.
> A point near one valid mode can still be penalized for being far from samples in another mode, so the resulting prediction set may be unnecessarily large by including low-density regions between modes.
> More importantly from the prediction-set perspective, after calibration one still needs to characterize $\{y:s(x,y)\le Q_{1-\alpha}\}$, which is generally implicit like in the case of MMD.
> This gives a practical reason for our current GMM-based method: with the dominant-component approximation, the resulting prediction set is an explicit union of ellipsoids,
> which is easier to interpret and use in downstream tasks.
>
>
> **Regarding the connection between Equations (5b) and (6).**
> We thank the reviewer for pointing it out.
> We agree that the original explanation was too intuitive.
> In the revision, we will add more explicit derivation showing that both equations use the same dominant-component negative log-mixture score, but with different density fits.
> Equation (5b) corresponds to a KDE-like mixture with $M$ isotropic components centered at each generated samples,
> while Equation (6) uses a $K$-component clustered Gaussian-mixture fit with learned weights, means, and covariances.
> We will also show that, for any fixed mixture, the log-sum score and log-max score difference is tightly upper bounded,
> and the gap becomes small when one mixture component dominates.
> Under the PCP specialization, the dominant-component negative log-density is rank-equivalent to the minimum Euclidean distance score used by PCP.
> Equation (6) then applies the same scoring template to a different, lower-complexity density estimate,
> leading to a prediction set given by a union of $K$ ellipsoids.

---

### Decision · Action_Editor_FQ8m · 2026-07-09

**Recommendation:** Accept with minor revision

**Additional Comments:**

There are several changes that the authors have proposed in their rebuttal. These should be incorporated into the final revision of the manuscript. The paper can be accepted provided that these revisions are satisfactorily implemented.

**Audience:**

Yes

**Audience Explanation:**

While reviewers find the novelity and impact of the paper to be rather limited, they agree that the paper meets the criterion related to "Audience".

**Claims And Evidence:**

Yes

**Claims Explanation:**

All reviewers agree that the paper clearly meets the criterion related to "Claims And Evidence" by providing claims that are sound and well supported by evidence. The reviewers raised some questions and concerns, which the authors have adequately addressed in the revisions.